# Opposing Roles of Blood-Borne Monocytes and Tissue-Resident Macrophages in Limbal Stem Cell Damage after Ocular Injury

**DOI:** 10.3390/cells12162089

**Published:** 2023-08-18

**Authors:** Chengxin Zhou, Fengyang Lei, Mirja Mittermaier, Bruce Ksander, Reza Dana, Claes H. Dohlman, Demetrios G. Vavvas, James Chodosh, Eleftherios I. Paschalis

**Affiliations:** 1Department of Ophthalmology, Massachusetts Eye and Ear and Schepens Eye Research Institute, Mass General Brigham, Harvard Medical School, Boston, MA 02114, USA; gevrrod@gmail.com (C.Z.); dfylei@icloud.com (F.L.); mirja.ramke@gmx.de (M.M.); bruce_ksander@meei.harvard.edu (B.K.); reza_dana@meei.harvard.edu (R.D.); claes_dohlman@meei.harvard.edu (C.H.D.); jchodosh@salud.unm.edu (J.C.); 2Boston Keratoprosthesis Laboratory, Department of Ophthalmology, Massachusetts Eye and Ear, Mass General Brigham, Harvard Medical School, Boston, MA 02114, USA; 3Disruptive Technology Laboratory, Department of Ophthalmology, Massachusetts Eye and Ear, Harvard Medical School, Boston, MA 02114, USA; 4Retina Service, Department of Ophthalmology, Massachusetts Eye and Ear, Mass General Brigham, Harvard Medical School, Boston, MA 02114, USA; demetrios_vavvas@meei.harvard.edu; 5Department of Ophthalmology and Visual Sciences, University of New Mexico School of Medicine, Albuquerque, NM 87108, USA

**Keywords:** limbal stem cell deficiency, cornea, epithelium, macrophages, monocytes, inflammation

## Abstract

Limbal stem cell (LSC) deficiency is a frequent and severe complication after chemical injury to the eye. Previous studies have assumed this is mediated directly by the caustic agent. Here we show that LSC damage occurs through immune cell mediators, even without direct injury to LSCs. In particular, pH elevation in the anterior chamber (AC) causes acute uveal stress, the release of inflammatory cytokines at the basal limbal tissue, and subsequent LSC damage and death. Peripheral C-C chemokine receptor type 2 positive/CX3C motif chemokine receptor 1 negative (CCR2^+^ CX3CR1^−^) monocytes are the key mediators of LSC damage through the upregulation of tumor necrosis factor-alpha (TNF-α) at the limbus. In contrast to peripherally derived monocytes, CX3CR1^+^ CCR2^−^ tissue-resident macrophages have a protective role, and their depletion prior to injury exacerbates LSC loss and increases LSC vulnerability to TNF-α-mediated apoptosis independently of CCR2^+^ cell infiltration into the tissue. Consistently, repopulation of the tissue by new resident macrophages not only restores the protective M2-like phenotype of macrophages but also suppresses LSC loss after exposure to inflammatory signals. These findings may have clinical implications in patients with LSC loss after chemical burns or due to other inflammatory conditions.

## 1. Introduction

Ocular chemical injury can lead to corneal blindness even if promptly treated [1]. One of the most important anatomic structures affected by the injury is the limbal basal layer [2], which contains a unique population of stem cells responsible for maintaining and replenishing the corneal epithelium [3]. Complete damage to the limbal niche results in limbal stem cell (LSC) deficiency, loss of corneal transparency, and subsequent vision loss [4]. LSC transplantation, using ex vivo reconstruction of autologous limbal epithelial cells (Holoclar^®^) or LSC allotransplantation, can help restore vision, but long-term results are hampered by a gradual loss of donor cells and corneal transparency [5,6]. Therefore, preserving viable LSCs can help restore the corneal epithelium and provide the required donor cells for subsequent expansion and auto-graft. The purpose of this paper is to explore alternative cellular, immunological, and biochemical mechanisms that cause limbal stem cell deficiency and potentially propose new therapies for LSC protection.

Until now, LSC damage after alkali injury has been attributed to direct damage by the caustic agent [7]. However, recently we have seen significant LSC loss occurring even in the absence of direct contact with the agent [8,9,10] and inhibition of tumor necrosis factor alpha (TNF-α) and vascular endothelial growth factor (VEGF) prevents LSC death and subsequent conjunctivalization and vascularization of the cornea after alkali injury [11]. Here we show that secondary inflammation is an alternative mechanism of LSC loss in the absence of direct damage by the caustic agent. Mechanistically, this process is instigated by rapid pH and oxygen changes in the anterior chamber that cause acute uveal stress and subsequent recruitment of inflammatory monocytes, especially at the basal limbal tissue. Recruited monocytes secrete tumor necrosis factor alpha (TNF-α), which initiates programmed cell death at the limbus. In contrast, tissue-resident macrophages are protective; in the setting of injury, tissue-resident macrophage depletion exacerbates monocyte infiltration, TNF-α release and subsequent LSC loss. Moreover, tissue-resident macrophage depletion augments LSC susceptibility to TNF-α-mediated apoptosis ex vivo. Interestingly, the repopulation of the corneal by new tissue-residue macrophages after depletion restores the protective phenotype of corneal macrophages and prevents LSC loss following injury. These findings shed light on the complex physiological and immunological mechanisms of LSC loss and highlight the important immunoregulatory role of tissue-resident macrophages and the detrimental role of peripheral monocytes in LSC preservation.

## 2. Materials and Methods

### 2.1. Mouse Model of Corneal Alkali Injury

All animal procedures were performed in accordance with the Association For Research in Vision and Ophthalmology Statement for the Use of Animals in Ophthalmic and Vision Research and the National Institutes of Health Guidance for the Care and Use of Laboratory Animals. The study was approved by the Animal Care Committee of Massachusetts, Eye and Ear. C57BL/6, Balb/c, B6.129P-*Cx3cr1^tm1Litt^*/J (CX3CR1^EGFP/EGFP^ Stock 005582), B6.129(Cg)-*Ccr2^tm2.1lfc^*/J (CCR2^RFP/RFP^ Stock 017586), B6.129S-*Tnfrsf1a^tm1Imx^ Tnfrsf1b^tm1Imx^*/J (TNFR1/2^−/−^ Stock 003243) mice were obtained from the Jackson Laboratory (Bar Harbor, ME). CX3CR1^+/EGFP^::CCR2^+/RFP^ reporter mice were generated by in-house breeding [9,12,13,14]. Mice between the ages of 6–12 weeks were used for this study.

Corneal alkali injuries were performed as per our published protocol [9]. Briefly, mice were anesthetized using ketamine (60 mg/kg) and xylazine (6 mg/kg). Proparacaine hydrochloride USP 0.5% (Bausch and Lomb, Tampa, FL, USA) eye drop was applied to the cornea 1 min before the burn. The cornea was carefully dried with a Weck-Cel (Beaver Visitec International, Inc., Waltham, MA, USA). A 2 mm diameter filter paper soaked into 1 M sodium hydroxide (NaOH) solution was applied to the central cornea for 20 s, avoiding exposure of the peripheral cornea and limbal area. After the injury, the eye was promptly irrigated with sterile saline for 15 min. Buprenorphine hydrochloride (0.05 mg/kg) (Buprenex Injectable, Reckitt Benckiser Healthcare Ltd., Berkshire, United Kingdom) was administered subcutaneously for pain alleviation, and Polytrim eyedrops (polymyxin B/trimethoprim, Bausch & Lomb Inc., Bridgewater, NJ, USA) were applied to the eye to prevent bacterial infection. Mice were kept on a heating pad until fully awake.

### 2.2. Rabbit Model of Chemical Injury

Six Dutch-Belted female rabbits (Covance, Dedham, MA, USA) between 2 and 2.5 kg were anesthetized by intramuscular injection of ketamine hydrochloride INJ, USP (35 mg/kg), (KetaVed^®^, VEDCO, St. Joseph, MO, USA) and xylazine (5 mg/kg), (AnaSed^®^, LLOYD, Shenandoah, IA, USA). A topical anesthetic (0.5% proparacaine hydrochloride, Bausch & Lomb, Tampa, FL, USA) was applied to the operative eye. Corneal alkali injuries were performed using an 8 mm diameter filter paper soaked in 2N NaOH and applied to the central cornea for 10 s, followed by immediate eye irrigation with saline solution for 15 min. After the burn, yohimbine (0.1 mg/kg), (Yobine^®^, LLOYD, Shenandoah, IA, USA) was administered in a marginal ear vein to reverse anesthesia. A single dose of buprenorphine (0.03 mg/kg) (Buprenex Injectable, Reckitt Benckiser Healthcare Ltd., Berkshire, United Kingdom) was injected subcutaneously, and a transdermal fentanyl patch (12 mcg/hour) (LTS Lohmann Therapy System, Corp, NJ, USA) was applied to the skin for 3 days to alleviate pain. Polytrim eyedrops (polymyxin B/trimethoprim, Bausch & Lomb Inc., Bridgewater, NJ, USA) were applied to the eye for 7 days to prevent bacterial infection.

### 2.3. Anterior Chamber pH Elevation with Cannulation

Anterior chamber pH elevation was achieved by replacing the physiological aqueous humor of mice with an equal volume of balanced salt solution (BSS) adjusted to pH 11.4. pH to match the pH of the anterior chamber of mice and rabbits observed 1 min after corneal surface injury with NaOH-soaked filter paper [9]. Cannulation was performed using a glass micropipette needle fitted on a gas-powered microinjection system (MDI, South Plainfield, NJ, USA). The needle was mounted on a 3D stereotaxis device, observed under an ophthalmic surgical microscope (Carl Zeiss Meditec, Thornwood, NY, USA) and inserted into the anterior chamber (AC) through the clear cornea. A volume of 4 μL of aqueous humor was replaced by an equal volume of BSS pH 11.4. After aqueous humor replacement, the needle was kept in the AC for 5 min to prevent a backward leak through the wound. After the removal of the needle, the wound was spontaneously sealed by stromal edema. The site of injection was gently irrigated with normal sterile saline, and the eye was tapped with a cotton tip to confirm wound closure.

### 2.4. Monocyte Fate Mapping Using a Chimera

C57BL/6J mice were myelodepleted as previously described [12,13,14,15], with 3 intraperitoneal injections of busulfan (Sigma-Aldrich, St. Louis, MO, USA) (35 mg/kg) 7, 5, and 3 days prior to bone marrow transfer. CX3CR1^+/EGFP^::CCR2^+/RFP^ bone marrow cells (5 × 10^6^ total bone marrow cells) were transferred to myelodepleted C57BL/6J mice via tail injection 1 month prior to corneal alkali injury. Bactrim (trimethoprim 80 mg and sulfamethoxazol 400 mg resuspension in 400 mL drinking water) was given ad libitum for 15 days post-busulfan treatment.

### 2.5. Tissue-Resident Macrophage Depletion Using CSF1R Inhibitor

The tissue-resident macrophages of the limbus (and cornea) were depleted with 3 weeks of administration of PLX5622 CSF1R inhibitor. The compound was provided by Plexxikon Inc. (Berkeley, CA, USA) and formulated in AIN-76A standard chow by Research Diets Inc. (New Brunswick, NJ, USA). Complete tissue-resident macrophage depletion was achieved with a dose of 1200 ppm, which was given to the mice for 3 weeks.

### 2.6. Ex Vivo Corneal Culture Experiments

Eyeballs were enucleated from age-matched C57BL/6J mice (6–8 weeks) that received either a 3-week diet of AIN-76A standard chow or a normal diet. Immediately after euthanasia, the eyeballs were enucleated and washed in a cold Keratinocyte serum-free medium (KSFM, 17005042, Gibco, Grand Island, VT, USA). The cornea and limbus were dissected from each eye, and the iris was removed. The tissue was gently washed in cold KSFM and cut into two halves with a surgical blade. Half explants were plated separately in 24-well plates, with epithelium facing upward. After removing excess media around the tissue, the explants were left on ice for 15 min to allow tissue adhesion to the culture plate surface. The attached explants were then cultured in KSFM + 10% fetal bovine serum (FBS) at 37 °C, 95% humidity, and 5% CO_2_ overnight. Culture media was then changed to KSFM+ 10% FBS +/− murine TNF-α (5 ng/mL, 315-01A, PeproTech, Rocky Hill, NJ, USA) the next day, and incubated for 6 h. For immunohistochemistry, tissues were fixed with 8% Paraformaldehyde solution for 20 min and frozen in Optimal Cutting Temperature (OCT) compound (Tissue-Tek 4583, Torrance, CA, USA) on dry ice. For flow cytometry analysis, tissues were digested in Collagenase solution for 1 h at 37 °C (Worthington Biochemical, Lakewood, NJ, USA). Cells were washed once and blocked by Fc Block (Clone: 2.4G2, BD Pharmingen) before staining using a panel of fluorescent-labeled antibodies. FACS was performed using BD LSR II, and data were analyzed using FlowJo (BD, v10.7).

### 2.7. Immunohistochemical Analysis

For the whole globes, eyes were fixed using 4% paraformaldehyde (PFA) for 1 h at room temperature and frozen in OCT compound on dry ice. Multiple sagittal sections (~10 μm in thickness) were obtained from the center and the periphery of the globe. Tissue sections were transferred to positive-charged glass slides Superfrost^®^ Plus 75 × 25 mm 1 mm thickness (VWR, Radnor, PA, USA) for further processing.

For corneal flat mounts, eyes were first fixed in 4% PFA solution for 1 h at +4 °C and tissues containing cornea, limbus and conjunctiva were surgically dissected, washed in phosphate buffer solution (PBS) with 0.1% Triton-X100 at +4 °C, and blocked in a buffer containing 5% bovine serum albumin and 0.3% Triton-X100 for 2 h at +4 °C. Tissues were incubated with appropriate antibodies overnight at +4 °C, washed in PBS and re-incubated with secondary antibodies for 2 h at room temperature. Tissues were washed again in PBS and transferred to positively charged glass slides (Superfrost^®^ Plus 75 × 25 mm, 1 mm thickness, VWR, Radnor, PA, USA) with the epithelium facing upwards for flat mount imaging. Four relaxing radial incisions were made to generate four flat tissue quadrants. UltraCruzTM mounting medium (sc-24941, Santa Cruz Biotechnology, Santa Cruz, CA, USA) was applied, and the tissues were covered by No.1 coverslips and nail polish. The following antibodies were used: Mouse monoclonal anti-TNF-α primary antibody pre-conjugated with FITC (Novus, NBP1-51502) 1:100; Goat polyclonal anti-K12 primary antibody (Santa Cruz, sc-17101) 1:100; Rabbit monoclonal anti-K13 primary antibody (Abcam, Cambridge, MA, USA, ab92551) 1:200; Mouse monoclonal ABCB5 antibody (M3C2 1:100; donated from Drs. Markus Frank and Bruce Ksander).

### 2.8. In Situ Hybridization

Tissue cryosections were fixed in 4% PFA for 1 h at +4 °C. The slides were serially dehydrated with 50%, 70% and 100% ethanol. Tissue sections were treated with RNAscope (ACD, Newark, CA, USA) hydrogen peroxide for 10 min, followed by 20 min of RNAscope protease IV treatment (Cat. No 322336). Tissue sections were then incubated with different RNAscope probes, including target gene probe, negative control probe, and Mm-Ppib positive control probe (ACD, Newark, CA, USA) at 40 °C in a humidified HybEZ oven for 2 h. 6-step mRNA signal amplification was performed using the RNAscope 2.5 HD Red detection kit, according to the manufacturer’s protocol. Slides were washed twice in RNAscope wash buffer after each amplification step. mRNA signal was detected with a red chromogenic reagent. Tissue sections were counterstained with 50% Hematoxylin solution, and mounted in EcoMount medium (Biocare, Concord, CA, USA) with glass coverslips for image quantification.

### 2.9. Microscopy

Tissue slides were imaged using the Zeiss Axio Imager M2 (Zeiss, Germany) fluorescence microscope with 20× dry and 63× oil immersion objective lenses. Flat-mount tissues were imaged using Leica SP8 confocal microscope (Leica, Microsystems Inc., Buffalo Grove, IL, USA). Images were taken using 10×, 20×, 40× and 63× objective lenses and by performing z- axial scanning of 0.7, 0.6, 0.4 and 0.3 μm step size, respectively. ImageJ was used to obtain maximum and average projections of the *z*-axis image stacks. Amira (Thermo Scientific™, Hillsboro, OR, USA), 3-D rendering software ImageJ (Fiji version 2.0.-rc-54/1.51g) was used to generate volumetric images and color-coded depth maps. Corneal tissue-resident macrophages were quantified using z-stack projections and volumetric 3-D analysis.

### 2.10. Flow Cytometry

Tissues were harvested at different time points and treated with a collagenase Type I dissociation system (Worthington, Lakewood, NJ, USA) for subsequent flow cytometry. Different fluorescent antibodies against CD45 (Clone: 104), CD11b (Clone: M1/70), I-A/I-E (MHC-II) (Clone: M5/114.15.2), C-X_3_-C motif chemokine receptor 1 (CX3CR1; Clone: SA011F11), CD86 (clone: GL-1), CD206 (clone: C068C2), (Biolegend, San Diego, CA, USA) were used to identify cell markers. Approximately 6% of the corneal cells were analyzed using the BD LSR II cytometer (BD Biosciences, San Jose, CA, USA) and flow data were processed using the FlowJo software version 10.6.2 (Tree Star, Ashland, OR, USA). CX3CR1^+^ and C-C motif chemokine receptor 2 positive (CCR2^+^) cells in CX3CR1^+/EGFP^::CCR2^+/RFP^ double reporter or chimeras were quantified using endogenous EGFP^+^ and RFP^+^ expression, respectively.

### 2.11. TUNEL Labeling and Quantitation of DNA Fragmentation

TUNEL labeling was performed in tissue sections using the Roche TUNEL kit (Roche, 12156792910) according to the manufacturer’s protocol. A mounting medium with DAPI (UltraCruzTM, Santa Cruz Biotechnology, Dallas, USA, sc-24941) was placed over the tissue, followed by a coverslip. Tile images were taken with an epifluorescent microscope (Zeiss Axio Imager M2, Zeiss, Germany). DAPI signal (blue) was overlayed with Texas Red (TUNEL^+^ cells) and quantified using ImageJ to assess the number of TUNEL^+^ cells overlapping with DAPI in the areas of interest.

### 2.12. Image Quantification

Quantification of protein expression in tissue sections with immunofluorescent staining was performed using ImageJ software [16] (Fiji version 2.0.-rc-54/1.51g). The area of interest was circumscribed using the freehand selection tool, and the image’s red, green, and blue channels were separated. Quantification was performed according to the color of the portion of interest and quantified as pixel density. The outcomes were normalized to the DAPI stained area of the corresponding image, and results were presented as percentages (Antibody/DAPI % density). For each analysis, 3–5 tissue sections from a minimum of three animals were used. mRNA quantification using RNAscope assay was performed by manually counting the mRNA events using ImageJ, as suggested by the manufacturer. Each punctuated red dot in the image represents one mRNA copy. For each experimental group, 3–6 sections from three animals were quantified.

### 2.13. Statistics

Quantitative results were reported as means −/+ standard deviations. The normality of the data was assessed by the Shapiro–Wilk test. Depending on the normality, either Student’s *t*-test or Mann–Whitney U-test was performed to compare the means between the anti-TNF-α group and the control group. One-way and two-way ANOVA were performed in data sets containing multiple variables, followed by the Holm–Šídák pairwise multiple comparison correction tests. Analyses were performed using R Studio (Boston, MA, USA). Statistical significance was set at *p* < 0.05 (two-tail).

## 3. Results

### 3.1. Limbal Stem Cell Damage Occurs Even in the Absence of Direct Injury by Caustic Agent

To investigate whether LSC damage occurs even in the absence of direct injury to the limbus, we created central corneal alkali injuries in rabbits and mice. In this model, the peripheral cornea and limbus remain unaffected by the caustic agent (Figure 1A). Central corneal alkali injury in the rabbit eye caused significant cell death beyond the affected area, which extended to the peripheral cornea and limbus (Figure 1B–D). There was evidence for ongoing cell death by apoptosis in both the limbal epithelial and stromal cells, even 2 weeks after the injury, as indicated by terminal deoxynucleotidyl transferase dUTP nick end labeling (TUNEL) assay (Figure 1B,D), and as compared to the naive rabbit eyes (Figure 1C,D). The damage to the limbus resulted in extensive neovascularization and conjunctivalization of the corneal surface within 2 weeks and dilation of the limbal vessels (Figure 1E,F). Moreover, similar results were obtained from the mouse (Figure 1G–O).

### 3.2. Anterior Chamber pH Elevation Is an Alternative Mediator of Limbal Stem Cell Damage

Previous studies have shown that corneal alkali injury causes rapid aqueous humor pH elevation from 7.4 to 11.4 within seconds, resulting in inflammation of the uvea [9]. To investigate the contribution of aqueous humor pH elevation to LSC loss independently of ocular surface injury, we performed a controlled elevation of the aqueous humor pH by the anterior chamber (AC) cannulation of 4 μL of balanced salt solution (BSS) adjusted to a pH 11.4.

Cannulation of the AC with 11.4 pH caused severe neovascularization and conjunctivalization of the cornea, iris pigment dispersion and deposition onto the posterior surface of the cornea, and pupil margin deformation (Figure 2A). Cannulated eyes with 11.4 pH solution exhibited complete loss of ATP-binding cassette sub-family B member 5 (ABCB5^+^) expression, a marker of limbal epithelial stem cells, Ref. [17] by 3 months, as assessed by confocal microscopy (Figure 2B,C,G), suggesting a complete loss of limbal epithelial stem cells. In contrast, eyes cannulated with normal (7.4) pH solution did not exhibit pathologic changes in the AC or the iris, and the cornea remained avascular and transparent as assessed one month after the procedure (Figure 2D). As expected, cannulated eyes with 7.4 pH solution retained a normal population of ABCB5^+^ limbal epithelial stem cells at the limbus, as assessed at 3 months (Figure 2E,F,H). This model did not cause chemical injury to the corneal surface. A representative photo of the cannulation procedure is shown in Figure 2G.

### 3.3. Peripheral Inflammatory Monocytes Are Key Mediators of Limbal Stem Cell Damage

The role of inflammatory monocytes in LSC damage was evaluated using the double reporter CX3C motif chemokine receptor 1^Enhanced green fluorescent protein^::C-C chemokine receptor type 2^Red fluorescent protein^ (CX3CR1^+/EGFP^::CCR2^+/RFP^) mouse model [12] by in situ hybridization and protein tissue analysis. Typically, interstitial and tissue-resident macrophages express CX3CR1, while circulating inflammatory monocytes express CCR2 [12,13].

Within 24 h of a central corneal alkali injury, a significant number (17.4%) of CD45^+^ cells infiltrated into the corneal/limbus (Figure 3A,B). This was associated with an increase in the number of CCR2^+^ cells from 0.04% to 5.3% and of CX3CR1^+^ cells from 0.06% to 1.3% within 24 h (Figure 3B,C). By day 7, the number of the CCR2^+^ cells was disproportionally higher compared to the CX3CR1^+^ tissue-resident macrophages in the injured cornea, as shown by representative confocal images and quantification of the CX3CR1^+/EGFP^ and CCR2^+/RFP^ cells in the cornea/limbus (Figure 3D–F). In naive mice, the cornea was predominantly occupied by ramified CX3CR1^+/EGFP^ cells with scant CCR2^+/RFP^ cells present across the corneal tissue (Figure 3D,F). Seven days after the alkali injury, the number of CCR2^+/RFP^ cells was significantly higher compared to naive eyes (Figure 3E,F) with a high number of CX3CR1^+/EGFP^ cells located primarily at the limbal area (Figure 2E,F). Quantification of the two immune cell populations in flat mounts revealed a disproportional increase in CCR2^+/RFP^ cells relative to CX3CR1^+/EGFP^ cells (Figure 3F). The abnormal influx of inflammatory cells in the limbus was also observed in rabbit eyes after central corneal burn, characterized by CD45^+^ myeloid cell infiltration in the basal limbal tissue, which remained present even 3 months after the injury (Figure 3G–I).

Previous studies have revealed that TNF-α is upregulated after chemical injury [8,9,18]. By using flow cytometry were quantified the contribution of CX3CR1^+^ tissue-resident macrophages and CCR2^+^ peripheral monocytes in TNF-α release 24 h of the injury. Infiltrating CCR2^+^ CX3CR1^−^ monocytes contributed significantly to the TNF-α secretion in the tissue, as compared to the CX3CR1^+^ CCR2^−^ tissue-resident macrophages, which showed minimal contribution (Figure 4A–C). By using in situ hybridization and protein analysis, we showed that TNF-α mRNA and protein expression was significantly upregulated at the site of injury (central cornea) and in the basal limbal tissue harboring the LSCs, but not in the intervening peripheral cornea (Figure 4D–M).

### 3.4. Tissue-Resident Macrophages Are Key Regulators of Limbal Stem Cell Injury

To understand the differential role of peripheral and tissue-resident monocytes/macrophages in LSC damage, we employed a bone marrow CX3CR1^+/EGFP^::CCR2^+/RFP^ chimera model to fate map infiltrating monocytes/macrophages in the cornea and to study their specific contribution in LSC damage [13,19,20].

In naive bone marrow transferred (BMT) CX3CR1^+/EGFP^::CCR2^+/RFP^ mice, the cornea was predominantly populated by CX3CR1^+^ tissue-resident macrophages, with sparse blood-derive CCR2^+^ monocytes distributed around the limbus (Figure 5A,F). Corneal alkali injury in BMT mice triggered CCR2^+/RFP^ cell infiltration from the blood into the cornea and limbus (Figure 5B,F ‘Injured’), which resulted in significant limbal epithelial cells apoptosis within 24 h (Figure 5H,L). To understand the role of CX3CR1^+/EGFP^ tissue-resident macrophages in limbal stem cell damage, BMT mice at steady-state were treated with colony-stimulating factor 1 receptor (CSF1R) inhibitor (PLX5622) to eliminate tissue-resident CX3CR1^+/EGFP^ macrophages (Figure 5C–F). Indeed, CSF1R inhibition was able to deplete all tissue-resident macrophages of the cornea, confirming previous studies in tissue-resident macrophages of other tissues [13,14,19,21]. However, CSF1R inhibition by PLX5622 (PLX) did not affect the number of peripheral CCR2^+^ that infiltrated the cornea and limbus within 24 h of the injury (‘PLX + Injury’), (Figure 5C,F). Importantly, the depletion of tissue-resident macrophages exacerbated limbal epithelial cell apoptosis after injury, as compared to injured mice with intact tissue-resident macrophages (Figure 5H,I,L). Further studies showed that this effect was not attributed to the CSF1R inhibitor, which in the absence of ocular injury, did not cause limbal epithelial cell apoptosis or recruitment of peripheral CCR2^+^ monocyte into the tissue (Figure 5D,J,L).

To further assess the protective role of tissue-resident macrophages, we performed additional experiments in which macrophage-depleted mice were allowed to achieve tissue-resident macrophage repopulation by stopping the administration of CSF1R inhibitor for 2 months (PLX off (2m)). Cessation of CSF1R inhibitor re-established normal tissue-resident macrophages in the cornea and restored their protective role against limbal epithelial cell apoptosis (Figure 5E,K,L).

### 3.5. Depletion of Tissue-Resident Macrophages Sensitizes LSCs to TNF-α-Mediated Damage

To further understand the role of tissue-resident macrophages in suppressing immune-mediated LSC damage, corneal tissues from wild-type mice that were treated with CSF1R inhibitor for 3 weeks were explanted and co-cultured for 6 h in KSFM + 10% FBS medium supplemented with 5 ng/mL murine recombinant TNF-α. Macrophage depletion increased the susceptibility of the limbal epithelial cell to TNF-α-induced apoptosis from 3.7% (wild-type mice) to 32% (PLX-treated mice), (Figure 5M,N), suggesting that tissue-resident macrophages are important modulators of LSC susceptibility to TNF-α. This hypersensitivity of limbal epithelial cells to apoptosis after depletion of tissue-resident macrophages was not attributed to ex vivo culture condition or to primary cytotoxicity of the recombinant TNF-α (rTNF-α), as both were well tolerated and caused minimal or no apoptosis to the limbus in mice with intact macrophages (Figure 5M,N). Instead, it was attributed to the loss of the protective/reparative function of tissue-resident macrophages and their implication in regulating inflammation. Indeed, flow cytometric analysis of explanted corneal/limbal tissues demonstrated a switch in the polarization of tissue-resident macrophages from classically activated, M1 pro-inflammatory phenotype to an alternatively activated M2 anti-inflammatory phenotype upon exposure to rTNF-α ex vivo, as defined by the down-regulation of major histocompatibility complex Class II molecules (MHC-II^hi^) and CD86^hi^ and the upregulation of CD206^hi^ surface markers (Figure 6A,B,G, CTRL & CTRL + rTNF-α). This phenotype was abolished in mice theater with CSF1R inhibitor for 3 weeks in vivo prior to exposure to rTNF-α ex vivo, (Figure 6C,D,G, PLX & PLX + rTNF-α). Interestingly, cessation of the inhibitor for 2 months led to repopulation of the cornea with new tissue-resident macrophages that re-established the macrophages’ M2-like polarization upon exposure to rTNF-α ex vivo (Figure 6E,F,G, PLX OFF 2M, PLX OFF 2M + rTNF-α).

## 4. Discussion

Limbal stem cell deficiency is a common and frequent complication after ocular chemical burns. Previous studies assumed that LSC loss is caused either by direct damage from the caustic agent or from limbal ischemia due to vascular damage, or both [22]. Here we show that LSC loss is not mediated exclusively by these two mechanisms but also through physiological changes within the anterior chamber that cause an immune response in the uvea. We demonstrate that rapid pH elevation in the aqueous humor cause acute uveal stress and subsequent infiltration of peripheral monocytes into the basal limbal tissue (Figure 7). In contrast, tissue-resident macrophages suppress inflammatory damage to the LSCs, whereas their depletion exacerbates LSC loss.

In previous studies, we have shown that corneal alkali burns cause a rapid increase in anterior chamber pH from 7.4 to 11.5 within seconds [9]. Here we expand on this finding and show that direct AC pH elevation, without an ocular surface burn, contributes to LSC damage independent of the physical damage to the ocular surface. Although cells can survive such transient pH elevation [23], subsequent stress to the uvea activates an inflammatory response that damages the LSCs. Key cellular mediators in this process are the peripheral CCR2^+^ monocytes, that infiltrate the tissue, especially around the limbus. This process is independent of the actual physical damage to the surface; artificial elevation of the aqueous humor pH with cannulation causes a similar immune activation, LSC loss, and conjunctivalization of the cornea. To our knowledge, this is the first study to demonstrate that pH elevation in the anterior chamber is an important cause of LSC loss in ocular surface alkali injury.

The limbal niche is supported by the presence of a distinct vasculature with radially oriented hairpin loops of arteries and veins [24], derived from the palisades of Vogt. These vessels provide the stem cells with nutrition and various blood-borne substances that support the LSCs [25,26]. Owing to the distinct anatomical location of the limbus, LSCs are exposed to interactions with immune cells and are therefore susceptible to damage induced by inflammatory mediators [27,28]. Our results suggest that resident macrophages protect LSCs from destructive infiltrating cells or secretion of inflammatory mediators, but in extreme situations of stress, this protective effect is overwhelmed by the vast infiltration of immune cells and the production of inflammatory cytokines at the limbal area. Although we demonstrated the opposing roles of tissue-resident macrophages and infiltrated peripheral monocytes in LSC survival, further studies are required to fully characterize the immunological profiles of the two immune cell populations.

Previous studies have shown that tissue-resident macrophages have an anti-inflammatory/tissue-repair function during tissue damage [29,30] as compared to blood-borne CCR2^+^ monocytes, which are more pro-inflammatory and contribute to tissue damage [12,14,31,32,33,34,35,36]. Here we show that depletion of the tissue-resident macrophages increased LSC susceptibility to apoptosis following exposure to recombinant TNF-α and repopulation of the cornea by new tissue-resident macrophages after depletion restores the protective function and ameliorates LSC loss. This is corroborated by data showing that tissue-resident macrophages contribute minimally to TNF-α release, as compared to CCR2^+^ cells from the periphery, and following exposure to TNF-α, tissue-resident macrophages switch from classical to alternative activation. Since short (1 week) [20] or long-term (3 weeks) [19] inhibition of CSF1R can affect a variety of other immune cells [19,20,37], we employed an ex vivo model to study the role of tissue-resident macrophages in the absence of peripheral immune cell infiltration. Our experiments suggest that tissue-resident macrophages undergo a phenotype switch following exposure to TNF-α from classically activated M1 pro-inflammatory to alternatively activated M2 anti-inflammatory, as defined by the down-regulation of MHC-II^hi^ and CD86^hi^ and the upregulation of CD206^hi^ surface markers. Interestingly, this phenotype is abolished in mice treated with CSF1R inhibitor; however, cessation of the inhibitor for two months allows repopulation of the cornea with new tissue-resident macrophages and restores their ability to polarize to M2-like phenotype upon exposure to rTNF-α ex vivo. This suggests that tissue-resident macrophages of the cornea can be repopulated by the blood to provide important regulatory functions and tissue support. This finding aligns with recent publications showing that tissue-resident macrophages in other tissues play a protective rather than deleterious role [9,13,14,19,38]. This is an important finding supporting the important regulatory role of tissue-resident macrophages in corneal immunology and LSC survival. Conversely, we show that the peripherally infiltrating CCR2^+^ cells mediate the damage to the LSC, and as such, suppression of the CCR2 axis should be the therapeutic target.

## 5. Conclusions

In conclusion, preserving the corneal limbal epithelium after a severe chemical injury has important ramifications for patients [4,7,39,40,41,42,43,44]. Understanding and controlling the molecular and cellular mechanisms that cause LSC loss can minimize the damage to the ocular surface and improve the success rate of adjuvant therapies and surgical approaches. To this end, the CCR2 axis appears to drive the detrimental effects of LSC loss through TNF-α-mediated inflammation, while the tissue-resident CX3CR1 macrophages appear to have a protective and immunoregulatory role against LSC death. Taking advantage of these findings in a translational manner could help clinical outcomes of LSC deficiency after trauma or ocular surface inflammation. This study also provides new insights to potentially advance corneal cell therapy with the transplantation of autologous limbal stem cells [45], oral mucosal epithelial cells [46], or induced pluripotent stem cells [47], which depend on ocular surface inflammation. Lastly, these findings may help improve the clinical outcomes of penetrating keratoplasty, primarily in patients with multiple failed grafts, which are known to exhibit elevated risk of iatrogenic LSC deficiency from inflammation due to repeated surgical trauma [48]. Adequate immunomodulation based on the described mechanism could substantially improve clinical outcomes and prevent LSC deficiency in these clinical scenarios. Further clinical studies are warranted. 

## Figures and Tables

**Figure 1 cells-12-02089-f001:**
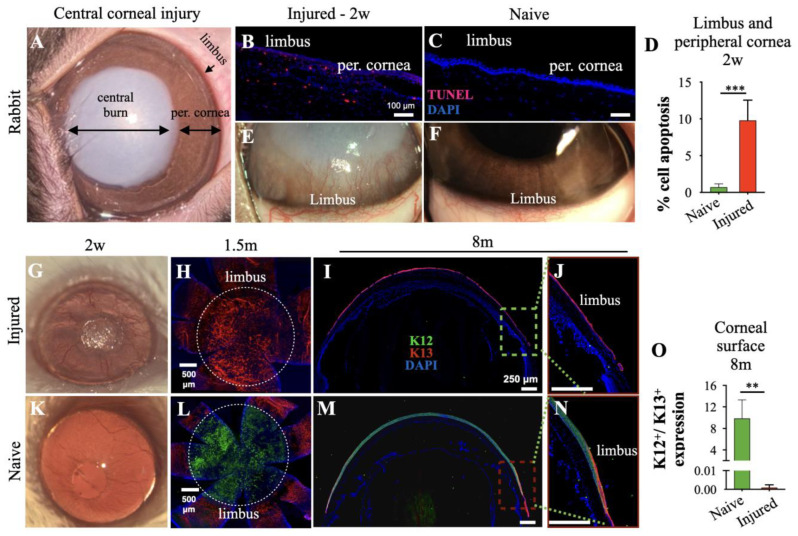
Limbal cell apoptosis triggered after central ocular alkali burn. (**A**) Biomicroscopic photo of a rabbit cornea immediately after 20 s of central 8 mm diameter alkali burn (NaOH 2N). (**B**,**C**) Limbal and peripheral corneal cell apoptosis 2 weeks after the injury using immunohistochemistry (IHC) and terminal deoxynucleotidyl transferase dUTP nick end labeling (TUNEL) assay. (**D**) Quantification of TUNEL^+^ cells. (**E**,**F**) Representative biomicroscopic photos of rabbit limbus showing severe neovascularization and opacification of the cornea 2 weeks after the injury. (**G**,**K**) Phenotypical appearance of C57BL/6 mouse eye 2 weeks after central corneal alkali burn. (**H**–**J**,**L**–**N**) IHC analysis of tissue sections and flat mounts of naive and alkali-injured mouse corneas using cytokeratine 12 (K12-green) and K13 (red) markers at 1.5 and 8 months post-injury. (**O**) Immunofluorescent quantification of K12^+^/K13^+^ expression 8 months after central corneal alkali burn. (**D**) n = 3 rabbits/group, *** *p* < 0.001, unpaired *t*-test, two-tailed, equal variance. (**O**) *n* = 4 mice/group, ** *p* < 0.01; unpaired *t*-test, two-tailed, equal variance. All data are expressed as means ± SD. K12—cytokeratin-12, K13—cytokeratin-13, per. cornea—peripheral cornea.

**Figure 2 cells-12-02089-f002:**
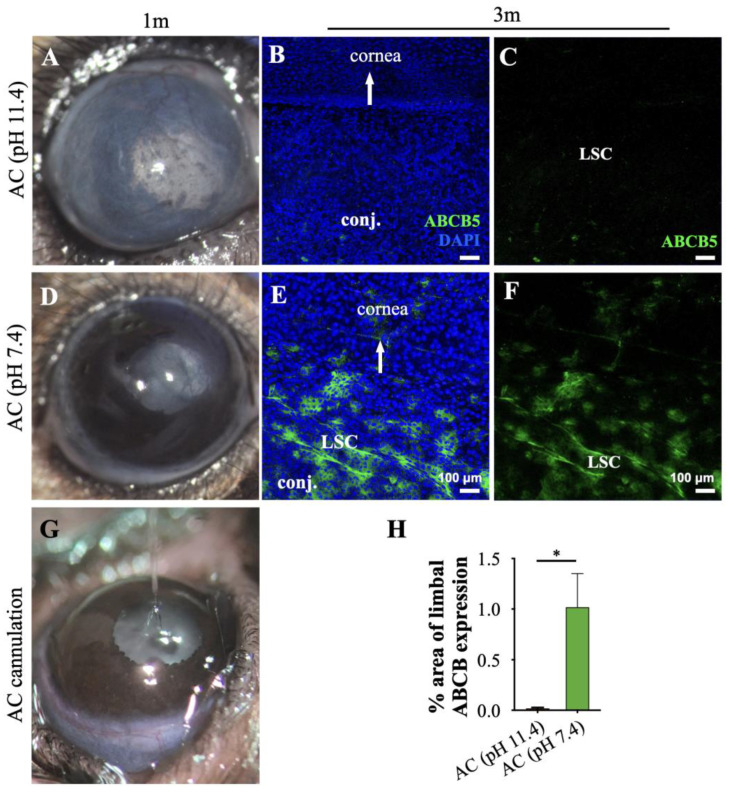
Anterior chamber pH elevation without ocular surface injury leads to LSC loss. (**A**,**D**) The phenotypical appearance of C57BL/6 mouse eye at one month after intracameral injection of 11.4 or 7.4 pH balanced salt solution (pH normalizes within 30–45 min) [9]. (**B**–**F**) Representative immunofluorescent staining (confocal maximum projection) of ABCB5 expressing limbal stem cells (green) of cannulated eyes at 3 months. (**G**) Representative photo of the anterior chamber cannulation procedure using a small glass capillary. (**H**) Quantification of ABCB5^+^ cells. *n* = 3 mice/group, * *p* < 0.05; unpaired *t*-test, two-tailed, equal variance. All data are expressed as means ± SD. LSC—limbal epithelial stem cell, AC—anterior chamber, Conj.—conjunctiva, ABCB5—ATP-binding cassette sub-family B member 5.

**Figure 3 cells-12-02089-f003:**
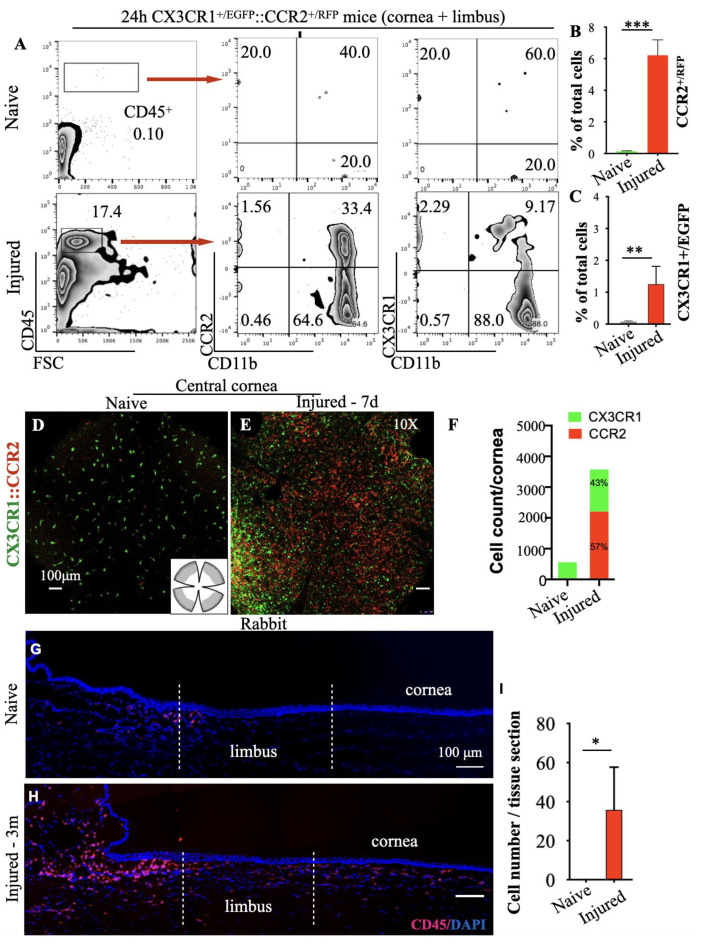
Blood-derived monocyte infiltration mediates tissue damage. (**A**–**C**) Representative flow cytometric plots and quantification of CD45^+^ cells from CX3CR1^+/EGFP^::CCR2^+/RFP^ reporter mice 24 h after corneal alkali injury. (**D**–**F**) Representative confocal images and quantification of CX3CR1^+/EGFP^ and CCR2^+/RFP^ cell infiltration in the cornea/limbus 7 days after corneal alkali injury in CX3CR1^+/EGFP^::CCR2^+/RFP^ reporter mice. (**G**–**I**) Immunohistochemistry (IHC) and quantification of cornea/limbus CD45^+^ cells 3 months after alkali injury in rabbit eyes. *n* = 3 animals/group, * *p* < 0.05, ** *p* < 0.01, *** *p* < 0.001, unpaired *t*-test, two-tailed, equal variance. CCR2—C-C chemokine receptor type 2, CX3CR1:CX3C chemokine receptor 1.

**Figure 4 cells-12-02089-f004:**
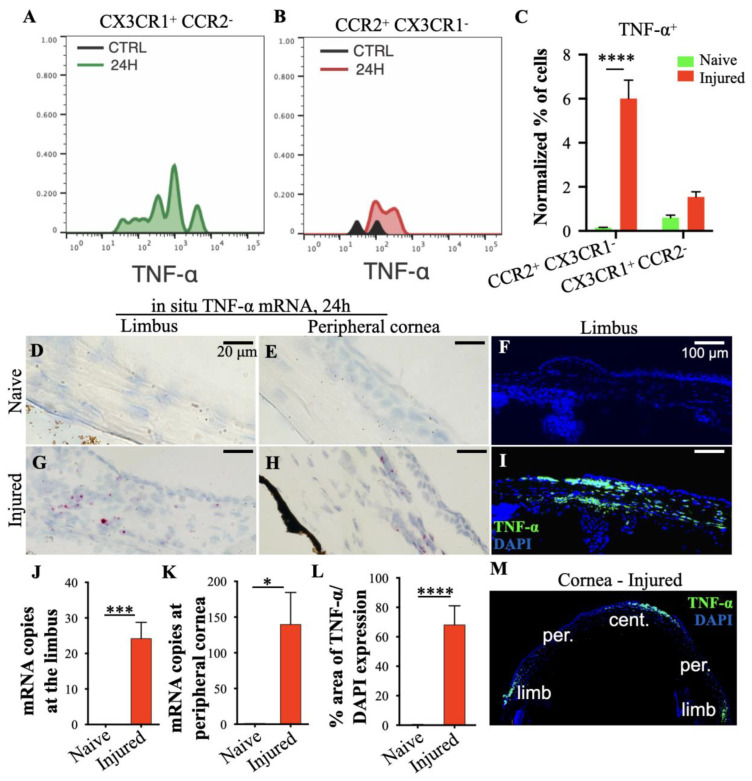
Limbal stem cell (LSC) damage is promoted by tumor necrosis factor alpha (TNF-α) upregulation in the basal limbal tissue. Animals received central corneal alkali burn (NaoH 1N) for 20 s [9], and tissues were collected and analyzed 24 h later. (**A**–**C**) Representative flow cytometry histograms and quantification of TNF-α expression from CCR2^+^ and CX3CR1^+^ cells. (**D**,**E**,**G**,**H**,**J**,**K**) Representative bright field images and quantification of the spatial distribution of TNF-α mRNA transcripts in the limbus and peripheral cornea of mice 24 h after injury using RNA-scope. Red dots represent TNF-α mRNA transcript. (**F**,**I**,**L**,**M**) Representative immunofluorescent images and quantification of TNF-α protein expression in limbus and cornea of mice 24 h after injury. (**C**) *n* = 3 mice/group, two-way ANOVA analysis with Šídák’s correction. (**J**–**L**) *n* = 3 animals/group, unpaired *t*-test, two-tailed, equal variance. All data are expressed as means ± SD. * *p* < 0.05, *** *p* < 0.001, **** *p* < 0.0001.

**Figure 5 cells-12-02089-f005:**
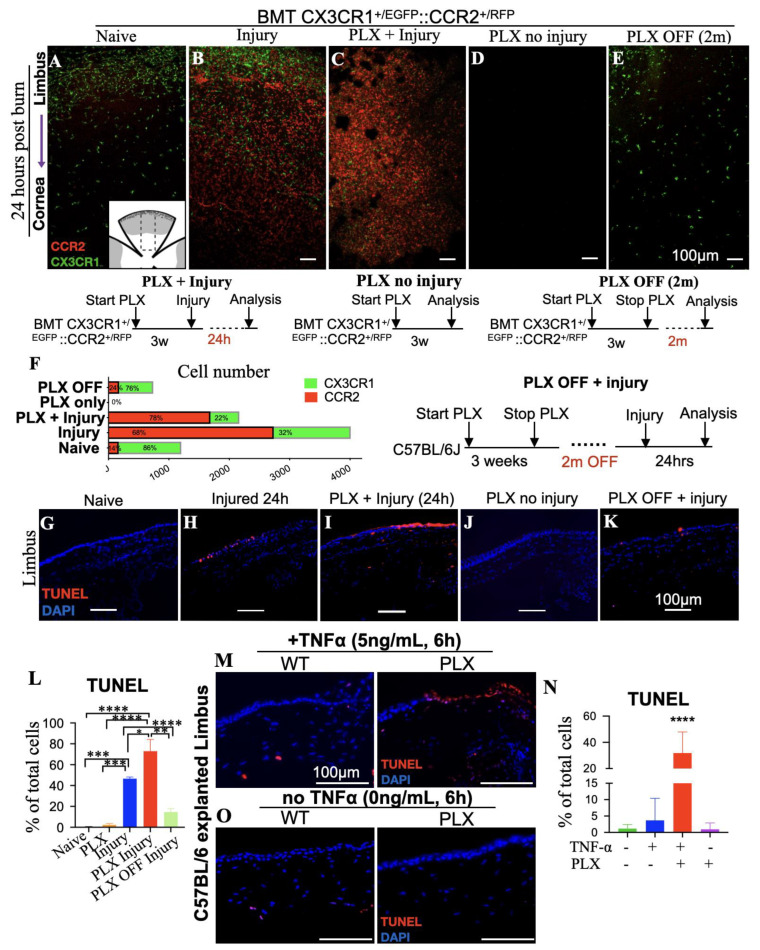
CCR2^+/RFP^ blood-borne monocytes, but not CX3CR1^+/EGFP^ resident macrophages, promote limbal cell death. Bone marrow-transferred animals received small-molecule CSF1R inhibitor PLX5622 for 3 weeks, followed by central cornea alkali burn with NaOH 1N. Tissues were collected and analyzed at various time points post-injury. (**A**–**F**) Representative flat mount confocal images and quantification of CX3CR1^+/EGFP^::CCR2^+/RFP^ bone marrow chimeras under different treatment conditions, including tissue-resident macrophage depletion using PLX5622 (PLX). (**G**–**L**) TUNEL assay and quantification of cell apoptosis 24 h after corneal alkali injury in wild-type mice under different treatment conditions, including PLX treatment. (**M**–**O**) Ex-vivo assessment of cornea-limbus cell apoptosis after incubation with 5 ng/mL murine TNF-α for 6 h in ex vivo cultures. Wild-type mice were either naive or pre-treated with PLX5622 for 3 weeks to deplete their tissue-resident macrophages. (**L**) *n* = 3 mice/group, one-way ANOVA with Tukey’s correction. (**N**) *n* = 5 animals/group, one-way ANOVA with Holm–Šídák’s correction. All data are expressed as means ± SD. * *p* < 0.05, ** *p* < 0.01, *** *p* < 0.001, **** *p* < 0.0001. BMT—bone marrow transfer, PLX—Plexxikon 5622.

**Figure 6 cells-12-02089-f006:**
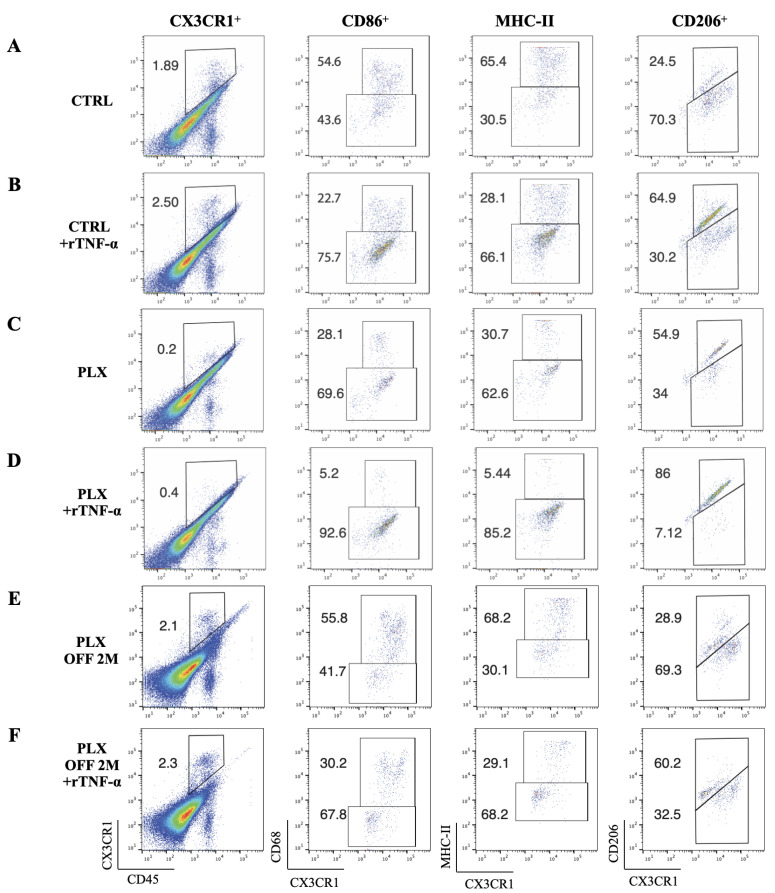
Tissue-resident macrophages in the cornea acquire a protective phenotype during inflammation. An ex vivo assessment of the phenotype of corneal tissue-resident macrophages following exposure to 5 ng/mL murine TNF-α for 6 h. (**A**–F) Representative flow cytometric plots and (**G**) quantification at 24 h and 2 months of classical and alternative polarization markers expressed by corneal tissue-resident macrophages following ex vivo exposure to recombinant TNF-α (rTNF-α) with and without exposure to the CSF1R inhibitor. (**G**) Exposure of explanted corneas to rTNF-α alters the polarization of CX3CR1^+^ tissue-resident macrophages from MHC-II^hi^ CD86^+^ CD206^−^ (classical activation) to MHC-II^lo^ CD86^−^ CD206^+^ (alternative activation). This phenotype switch is lost in mice treated with CSF1R inhibitor for macrophage depletion. Repopulation of tissue-resident macrophages after cessation of the CSF1R inhibitor for 2 months re-establishes the phenotype switch upon exposure to rTNF-α. *n* = 4 mice/group, * *p* < 0.05, **** *p* < 0.0001, one-way ANOVA, Dunnett’s multiple comparison test, all comparisons to CTRL group.

**Figure 7 cells-12-02089-f007:**
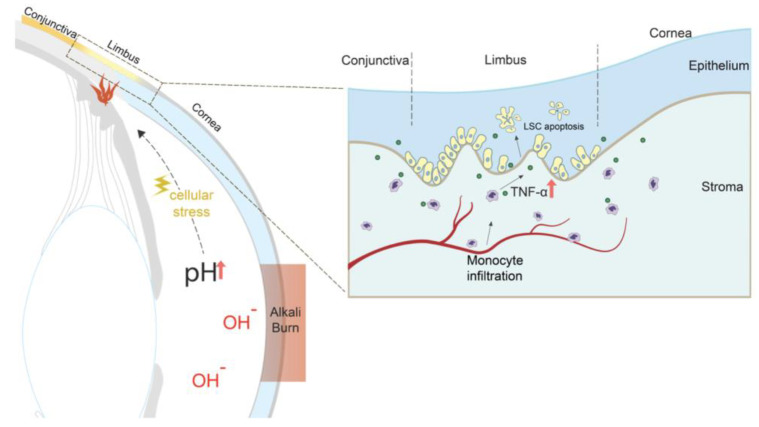
Proposed mechanism of limbal epithelial stem cell loss after central corneal alkali burn injury. Rapid pH elevation in the aqueous humor causes uveal stress and subsequent infiltration of peripheral monocytes into the basal limbal tissue; in contrast to tissue-resident macrophages, peripheral monocytes secrete a significant amount of TNF-α that causes limbal stem cell death. Prompt TNF-α inhibition after injury reduces monocyte infiltration into the limbus, suppresses inflammation, protects the limbal stem cell from apoptosis and preserves the ocular surface phenotype.

## Data Availability

Not applicable.

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
