# Peer review of "Opposing Roles of Blood-Borne Monocytes and Tissue-Resident Macrophages in Limbal Stem Cell Damage after Ocular Injury"

_cells, 2023, doi:10.3390/cells12162089_

Round 1
Reviewer 1 Report
LSC damage occurs through immune cell mediators without direct injury to LSCs. In this manuscript, the authors concluded that the CCR2 axis appears to drive the detrimental effects of LSC loss through TNF-α-mediated inflammation. The tissue-resident CX3CR1macrophages have a protective and immunoregulatory role against LSC apoptosis. This manuscript was well prepared.Author Response
Comments From Reviewer 1:
Reviewer 1
LSC damage occurs through immune cell mediators without direct injury to LSCs. In this manuscript, the authors concluded that the CCR2 axis appears to drive the detrimental effects of LSC loss through TNF-α-mediated inflammation. The tissue-resident CX3CR1macrophages have a protective and immunoregulatory role against LSC apoptosis. This manuscript was well prepared.
-We would like to thank the review for his time and effort to review our work and for his supportive comments..
Sincerely,
Eleftherios Paschalis Ilios
Reviewer 2 Report
Dear Authors,
Presented for your review is an important topic which is cellular mechanisms in stem cell injury of the cornea after eye injury. A thorough understanding of the mechanisms responsible for the injury is especially for researchers who specialize in cellular therapies and corneal stroma cell transplantation. To treat one must know the mechanisms of the disease. Currently, in many countries , due to the protection of laboratory animals, research on them is severely restricted.
I rate the quality of the research performed very highly.
Minor comments on the work.
1. please add in the supplement microscopic images in better quality 600 dpi.
2. figure 6b are illegible. Please think about redesigning figure 6.
3. the paper needs conclusions as a separate chapter, with reference to applications in regenerative medicine
4. in the introduction, please define the purpose of the paper. In my opinion, you should also refer to cell therapy methods, e.g. Holoclar, briefly.
Author Response
Comments From Reviewer 2:
Presented for your review is an important topic which is cellular mechanisms in stem cell injury of the cornea after eye injury. A thorough understanding of the mechanisms responsible for the injury is especially for researchers who specialize in cellular therapies and corneal stroma cell transplantation. To treat one must know the mechanisms of the disease. Currently, in many countries , due to the protection of laboratory animals, research on them is severely restricted.
I rate the quality of the research performed very highly. =
-We would like to thank the reviewer for his effort to improve our work and his supportive comments. We have revised the manuscript according to his suggestions. Thank you.
Minor comments on the work.
1. please add in the supplement microscopic images in better quality 600 dpi.
-Thank you. We replaced all images with higher quality.
2. figure 6b are illegible. Please think about redesigning figure 6.
-We redesigned Figure 6 to solve the problem. Thank you.
3. the paper needs conclusions as a separate chapter, with reference to applications in regenerative medicine.
-We added a Conclusion section and incorporated a statement about applications in regenerative medicine.
4. in the introduction, please define the purpose of the paper. In my opinion, you should also refer to cell therapy methods, e.g. Holoclar, briefly.
-We have revised the Introduction to state the purpose of the study (Line 58-61) and the relevance of this study in long-term success of various LSC therapies, including Holoclar (Line 50-58).
Reviewer 3 Report
Zhou and co-authors presented the opposing roles of blood-borne monocytes and tissue-resident macrophages in limbal stem cell damage after ocular injury. Their results showed the limbal stem cell damage after chemical injury to the eye is mediated by immune cell mediators, even without direct injury to the limbal stem cells. The study was well-designed, and the results strongly support the conclusion. This is a significant contribution to our understanding of corneal wound healing and this research field.
I have few minor comments as follow:
I recommend that all abbreviations used throughout the manuscript be defined in full at their first presentation. There are many abbreviations that are missing definitions.
The overall graphs are somewhat blurry and need improvement for quality, especially Fig 3A.
Add a scale bar to all histo/IHC images.
In Figure 6, the title line at the top should be removed. How about adding more numbers (A, B, C, D…) to each group, like the other figures? This would make the figure 6 more readable.
In line 287, the number “(23)” should be superscripted.
Author Response
Comments From Reviewer 3
Zhou and co-authors presented the opposing roles of blood-borne monocytes and tissue-resident macrophages in limbal stem cell damage after ocular injury. Their results showed the limbal stem cell damage after chemical injury to the eye is mediated by immune cell mediators, even without direct injury to the limbal stem cells. The study was well-designed, and the results strongly support the conclusion. This is a significant contribution to our understanding of corneal wound healing and this research field.
-We highly appreciate the comments from the reviewer and we have revised the paper accordingly. Thank you
I have few minor comments as follow:
I recommend that all abbreviations used throughout the manuscript be defined in full at their first presentation. There are many abbreviations that are missing definitions.
-Thank you for the suggestion. All abbreviations in the manuscript are now defined at first time.
The overall graphs are somewhat blurry and need improvement for quality, especially Fig 3A
-Thank you. The draft figures were compressed. High-resolution images will be included in the final submission.
Add a scale bar to all histo/IHC images.
-We have added scale bars to all IF and IHC images.
In Figure 6, the title line at the top should be removed. How about adding more numbers (A, B, C, D…) to each group, like the other figures? This would make the figure 6 more readable.
-We have removed the title line and added numbers to each experimental group.
In line 287, the number “(23)” should be superscripted.
-Thanks for noticing the error. We corrected the citation.
Thank you for your review and suggestions.
Sincerely,
Eleftherios Paschalis Ilios